# COVID-19 Pandemic-Adapted Radiotherapy Guidelines: Are They Really Followed?

Elena Galofaro [1,2,*], Claudio Malizia [3], Ilario Ammendolia [1], Andrea Galuppi [1], Alessandra Guido [1], Maria Ntreta [1], Giambattista Siepe [1], Giorgio Tolento [1], Antonio Veraldi [1], Erica Scirocco [1,2], Alessandra Arcelli [1,2], Milly Buwenge [1,2], Martina Ferioli [1,2], Alice Zamagni [1,2], Lidia Strigari [4], Silvia Cammelli [1,2,†] and Alessio Giuseppe Morganti [1,2,†]

1   Radiation Oncology, IRCCS Azienda Ospedaliero-Universitaria di Bologna, 40138 Bologna, Italy;
    ilario.ammendolia@aosp.bo.it (I.A.); andrea.galuppi@aosp.bo.it (A.G.); alessandra.guido@aosp.bo.it (A.G.);
    maria.ntreta@aosp.bo.it (M.N.); giambattista.siepe@gmail.com (G.S.); giorgio.tolento@aosp.bo.it (G.T.);
    antonio.veraldi@aosp.bo.it (A.V.); erica.scirocco@studio.unibo.it (E.S.); alessandra.arcelli2@unibo.it (A.A.);
    mbuwenge@gmail.com (M.B.); m.ferioli88@gmail.com (M.F.); alice.zamagni@yahoo.it (A.Z.);
    silvia.cammelli2@unibo.it (S.C.); alessio.morganti2@unibo.it (A.G.M.)
2   Department of Experimental, Diagnostic, and Specialty Medicine—DIMES, Alma Mater Studiorum Bologna
    University, 40138 Bologna, Italy
3   Nuclear Medicine, IRCCS Azienda Ospedaliero-Universitaria di Bologna, 40138 Bologna, Italy;
    claudio.malizia@aosp.bo.it
4   Medical Physics, IRCCS Azienda Ospedaliero-Universitaria di Bologna, 40138 Bologna, Italy;
    lidiastrigari4@gmail.com
*   Correspondence: elena.galofaro@studio.unibo.it; Tel.: +39-3409328999
†   Contributed equally.

**Abstract:** Background: In our department, we provided guidelines to the radiation oncologists (ROs) regarding the omission, delay, or shortening of radiotherapy (RT). The purpose was to reduce the patients' exposure to the hospital environment and to minimize the departmental overcrowding. The aim was to evaluate the ROs' compliance to these guidelines. Methods: ROs were asked to fill out a data collection form during patients' first visits in May and June 2020. The collected data included the ROs' age and gender, patient age and residence, RT purpose, treated tumor, the dose and fractionation that would have been prescribed, and RT changes. The chi-square test and binomial logistic regression were used to analyze the correlation between the treatment prescription and the collected parameters. Results: One hundred and twenty-six out of 205 prescribed treatments were included in this analysis. Treatment was modified in 61.1% of cases. More specifically, the treatment was omitted, delayed, or shortened in 7.9, 15.9, and 37.3% of patients, respectively. The number of delivered fractions was reduced by 27.9%. A statistically significant correlation ($p = 0.028$) between younger patients' age and lower treatment modifications rate was recorded. Conclusion: Our analysis showed a reasonably high compliance of ROs to the pandemic-adapted guidelines. The adopted strategy was effective in reducing the number of admissions to our department.

**Keywords:** radiotherapy; guidelines; compliance; COVID-19

## 1. Introduction

The severe acute respiratory syndrome coronavirus 2 (SARS-CoV-2) is highly contagious and transmitted from human to human via respiratory droplets or virus contaminated surfaces that causes the coronavirus disease 2019 (COVID-19) [1]. COVID-19 was identified in Wuhan (Hubei Province, China) in December 2019 and was recognized by the World Health Organization (WHO) as an international pandemic in May 2020. Due to the increasing number of infected patients worldwide, COVID-19 poses an unprecedented challenge to healthcare systems. In fact, the latter had to face the emergency in several ways, including the development of pandemic-adapted guidelines in different settings [2].

The COVID-19 pandemic also represents a complex situation for radiation oncologists (ROs) [3–5]. In particular, one of the main challenges is to reduce the departments' overcrowding by lowering the admission rate to radiotherapy (RT) departments. This attempt has two different purposes.

First, fewer visits to the RT center can reduce the risk of contagion for healthcare workers. This lowered risk, besides protecting the staff, minimizes the need for RT delays and interruptions. In fact, the hospitalization or quarantine of part of the staff would hinder RT planning and delivery with a detrimental impact on treatment effectiveness [6]. Second, reducing the departments' crowding decreases the contagion risk for patients undergoing RT. This is also particularly important given the higher mortality rate from COVID-19 in cancer patients [7].

For this purpose, several guidelines to reduce the infectious risk in RT departments have been published [8–16]. However, the publication and dissemination of guidelines does not lead to their automatic application/implementation. A well-known example in RT is that guidelines recommending a single fraction in the treatment of uncomplicated bone metastases are often disregarded [17–19]. Therefore, it would be useful to evaluate the effectiveness of guidelines adapted for emergency situations. In particular, it would be useful to analyze the actual compliance of ROs to new guidelines in order to optimize the management of the present and any future pandemics.

Some surveys have been published about prescriptive changes, as "declared" by the ROs, produced by the pandemic [20–22]. However, analyses on the real ROs' compliance to pandemic-adapted guidelines are lacking. Furthermore, information on the real impact of new guidelines on the admissions rate in RT departments is missing.

Therefore, the purpose of this analysis was to perform a prospective assessment of ROs' compliance to COVID-19-adapted guidelines. Moreover, our aim was to assess the influence of several parameters on the adherence to guidelines and their impact on the admission rate in our department.

## 2. Materials and Methods

This is a single-center prospective observational study approved by the Ethics Committee of our institution (CHRISTIE, CE: 989/2020/Oss/AOUBo). Patients signed a written informed consent for participation in the study prior to enrollment.

The primary end point of the study was the compliance rate to the new guidelines introduced at the beginning of May 2020 in our RT department. The new guidelines suggested, in some specific settings, the possibility of avoiding or postponing RT or using fewer fractions compared to the standard departmental guidelines. Therefore, the main objective of the analysis was to evaluate the percentage of patients for whom, according to the COVID-19-adapted guidelines, RT was omitted, delayed, delivered in a shorter time, or without changes. In addition, the overall reduction in admission in the RT department was calculated. Finally, the potential correlation between the following parameters and the new guidelines' compliance was investigated: RO's age and gender, patient's age and place of residence, treated tumor, and RT aim (curative vs. palliative).

All patients who underwent evaluation for RT between 1 May 2020 and 30 June 2020 were included in the study. However, patients younger than 18 years and those selected for particularly short treatments (palliative RT with single fractionation, accelerated palliative RT delivered in four fractions/two days, and stereotactic RT) were excluded.

In the first two months of the pandemic in Italy (March–April 2020), the following series of measures were taken in our department: (i) organization of a triage for patients and healthcare professionals (body temperature control, investigation of symptoms and contacts with SARS-CoV-2 positive subjects); (ii) distribution of personal protection systems of varying complexity to the staff depending on the operations to be performed; and (iii) remote follow-up visits, except for the first visit after treatment or in the case of symptomatic patients or patients who reported abnormal blood or instrumental tests.

At the end of April 2020, two published guidelines [15,23] proposing adaptations of RT to the pandemic situation were selected. These guidelines were chosen as they were relatively simple and included the majority of neoplasms. In a first meeting (24 April 2020), the guidelines were explained to all ROs in the department. In the following days, three authors (E.G., I.A., A.G.M.) prepared and provided the following to the ROs:

- A booklet summarizing the guidelines in Italian;
- A data collection form to be filled in at the time of the first visit, to record the data mentioned above;
- The patient's written informed consent form.

On 27 April 2020, a second meeting took place in which this documentation was distributed to the ROs to collect data from 1 May 2020 to 30 June 2020.

Data were analyzed using R version 3.5.0 statistic software (R Foundation for Statistical Computing, Vienna, Austria). A chi-square test was performed to evaluate the correlation between the analyzed variables (age and gender of the prescribing RO, patient age and residence, treatment intent, and tumor site) and prescription modifications. A binomial logistic stepwise regression was used to estimate the likelihood of prescription modifications based on the above-mentioned variables. A *p*-value of less than 0.05 was considered statistically significant.

## 3. Results

### 3.1. Patients Characteristics

Over the two-month period of our study, 205 patients were evaluated for RT. Of these, 79 patients were excluded for the following reasons: (i) RT not indicated (10); (ii) RT refused by the patient (3); (iii) lack of possible treatment changes in the new guidelines (39); (iv) treatment with stereotactic RT (16); (v) palliative single fraction RT (11). The characteristics of patients included in the study (126) are shown in Table 1.

**Table 1.** Chi-square analysis: correlation between prescription modification and analyzed variables.

| | | Treatment | | *p*-Value (Pearson Chi-Square) |
|---|---|---|---|---|
| | | Modified (%) | Non Modified (%) | |
| Radiation oncologist's age (years) | 30–40 | 81.5 | 18.5 | 0.004 |
| | 40–50 | 50.0 | 50.0 | |
| | 50–60 | 40.0 | 60.0 | |
| | >60 | 74.2 | 25.8 | |
| Radiation oncologist's gender | Male | 67.9 | 32.1 | 0.045 |
| | Female | 50.0 | 50.0 | |
| Patient's age (years) | 30–50 | 42.1 | 57.9 | 0.082 |
| | 50–60 | 52.4 | 47.6 | |
| | >60 | 67.4 | 32.6 | |
| Patient's residence | Same city or province of the department | 60.4 | 39.6 | 0.447 |
| | Same region of the department | 80.0 | 20.0 | |
| | Outside the region of the department | 57.1 | 42.9 | |
| Treated tumor | Prostate cancers | 66.7 | 33.3 | 0.412 |
| | Breast cancers | 64.9 | 35.1 | |
| | Other primary tumors | 45.5 | 54.5 | |
| | Metastases | 62.5 | 37.5 | |
| Treatment aim | Curative | 61.2 | 38.8 | 0.983 |
| | Palliative | 61.0 | 39.0 | |

### 3.2. Compliance to the Guidelines

Overall, RT was modified according to the new guidelines in 77 (61.1%) patients included in the study. In particular, RT was omitted in 10 (7.9%) patients: (i) adjuvant treatment was omitted in five patients considering the low risk of relapse; (ii) systemic therapy was prolonged in two patients; (iii) in three patients referred for palliative RT but with mild symptoms, the supportive care was adjusted.

Instead, RT was delayed in 20 (15.9%) patients. In 17 of these, with castration-sensitive prostate cancer, RT was temporarily replaced with androgen deprivation therapy. In addition, RT was postponed in two patients with basal cell carcinoma of the skin and in one patient with cerebral lymphoma and SARS-CoV-2-related pneumonia.

Finally, RT was delivered with fewer number of fractions compared to our departmental standards in 47 (37.3%) patients. In particular, (i) 22 patients with breast cancer underwent accelerated RT according to the FAST-Forward protocol [24]; (ii) 18 palliative treatments were delivered with fewer fractions than usual; (iii) two patients with a low–intermediate risk prostate cancer underwent stereotactic RT; (iv) fewer fractions were used in two patients with non-melanoma skin cancer, one patient with brain lymphoma, one with plasmacytoma, one with neuroendocrine carcinoma, and one patient with Merkel's carcinoma (Figure 1).

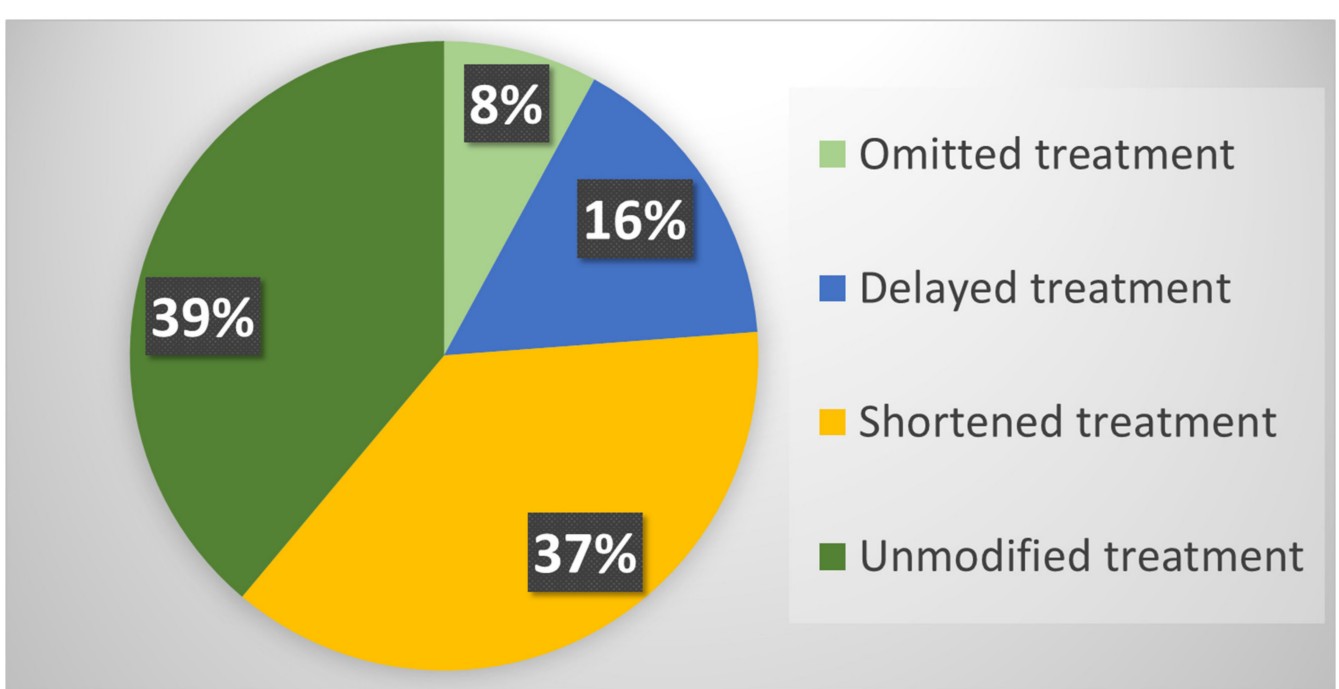

**Figure 1.** Distribution of treatment changes based on guidelines.

RT modifications were different between the different tumors. For example, in breast tumors, no treatment was delayed, while in 56.8% of cases, it was performed with a reduced number of fractions using the FAST-Forward regimen. On the contrary, in patients with prostate cancer, the number of fractions was reduced only in two (6.9%) cases by using stereotactic RT. Moreover, given the opportunity of prolonging neoadjuvant hormone therapy, RT was delayed in 55.2% of them.

Overall, RT was planned and delivered as per pre-COVID-19 guidelines in 49 (38.9%) patients. Considering all the patients evaluated during the study period, and therefore, also the subjects excluded from the analysis, the rate of modified RT was 37.6%. In addition, the introduction of adapted guidelines reduced the number of RT fractions delivered over the two-month period of the study by 27.9% (888: from 3182 expected based on the standard guidelines to 2294 actually delivered).

None of the healthcare, technical, and administrative components of the staff and of the treated patients suffered a SARS-CoV-2 infection in the same period.

### 3.3. Correlations with Compliance to the Guidelines

The results of the univariate analysis are shown in Table 1. Compliance to the guidelines was higher in younger (<40 years old) and older (>60 years old) ROs compared to the ones with intermediate age (40–60 years) ($p = 0.004$). Using the residual test, based on the difference between observed and expected values, this significant difference was mainly due to the higher tendency to adhere to guidelines of the younger compared to middle-aged ROs. Moreover, a higher compliance was noted in male ROs ($p = 0.045$). Finally, the chi-square test showed a trend for reduced compliance to the new guidelines in younger (30–50 years old) patients ($p = 0.082$). The multivariable analysis performed with a binomial logistic regression confirmed only a statistically significant correlation between older patients age and probability of compliance to the adapted guidelines ($p = 0.028$).

## 4. Discussion

The compliance to new guidelines adapted for the COVID-19 pandemic was analyzed in a prospective observational study. Although several guidelines have been published on RT during this pandemic period, to the best of our knowledge this is the only study analyzing the real compliance to the indications available in the literature.

Our results showed a reasonable adherence to the guidelines, with values close to two thirds of the patients included in the analysis and over one third of the general population of the patients who visited the department. These results can be considered positively considering the very short time (about one week) during which the new therapeutic indications were presented and suggested to all ROs.

The higher compliance in terms of adherence to the new guidelines in younger and older ROs that we observed in the chi-square test was based on the higher tendency to adhere to guidelines of the younger compared to middle-aged ROs. However, the low number of doctors in our department allowed us to describe what happened in our center but not to apply the results on a larger scale. The multivariate analysis confirmed that the adherence to the adapted guidelines was influenced only by the patient's age. In fact, in patients over 60 years of age, the compliance rate was 67.4%, while in patients aged less than 50 years, the compliance rate was 42.1%. This finding is quite understandable considering the expected reluctance to de-escalate treatment in younger patients and the higher mortality from COVID-19 in those over 60 years of age.

One could assume a greater adherence to guidelines for the patients residing farthest from the RT department, and in particular, a greater use of accelerated treatments to reduce the need for repeated movements with the related risk of infection. However, it should be emphasized that our department is located in a city (Bologna) at the center of the Italian communication networks (roads, rail, and air routes), and in particular has high-speed train connections with most of the other Italian regions. In addition, during the study period, various safety measures had been adopted in public transports, such as distancing between passengers, supply of face masks and disinfectants, and separate ways of entering and exiting in buses and trains. This may partially explain the lack of impact of the place of residence on guidelines compliance.

It is worth noting that the new guidelines were not applied in 38.5% of the patients included in the study. This was partly due to the lack of inclusion of some clinical settings in the published guidelines used in our experience. About 50.0% of these patients had gynecological malignancies referred for curative RT, for which there are no alternatives to standard fractionated and timely delivered treatment. Furthermore, we excluded the possibility of modifying RT in the case of short palliative treatments (1–2 days), considering both the need of symptoms relief and the minimal number of patients visits in the hospital, and stereotactic treatments, also in this case, considering the short treatment duration.

During the study period, 67 people worked in our RT department: ROs, residents, physicists, nurses, RT technicians, and administrative staff. Considering that our department is based in a university hospital that became a COVID-19 reference center during the pandemic, the absence of COVID-19 cases among both staff and patients can be considered a positive effect, to which the adoption of new guidelines could have contributed.

However, it should be emphasized that the adapted guidelines were only one of the adopted measures, together with (i) control of all the subjects (staff and patients) at the entrance to the department, with symptoms and body temperature checks; (ii) mandatory use of disposable gloves in the examination rooms, bunkers, and CT-simulator room, mandatory use of protective masks for staff and patients; (iii) interruption of all the clinical, organizational, and institutional meetings with the transition to online meetings and teaching; (iv) follow-up visits with telemedicine systems; (v) patients distancing in waiting rooms, access to the department only for the patient without accompanying family members (except in special cases of needed personal assistance), and the provision of personal protection systems and disinfectants in all the department spaces.

Our study has some obvious limitations. First, a rather small number of patients and staff members was included. This small sample size probably prevented the identification of other factors affecting the compliance to the guidelines beyond patients' age. Furthermore, the study was mono-centric and, therefore, probably influenced by the local traditional prescription habits.

Therefore, the conclusions of the analysis are only partially generalizable. Additionally, the timing of the study may have influenced the results. For example, only two days before the start of data collection (28 April 2020) the results of the FAST-Forward trial on the adjuvant treatment in only five fractions of breast cancer became available online [24]. The results of this important randomized trial probably stimulated the adoption of this regimen in ours as well as in other centers [25].

At the same time, the trend of the pandemic in the study period may have reduced the interest in the adoption of the new guidelines. In fact, in the Italian region where our department is based (Emilia-Romagna), the peak of SARS-CoV-2 infections was reached in mid-April 2020, and was followed by a progressive reduction, with a similar trend in the number of COVID-19-related deaths. If the analysis had been performed in the previous two months (March–April 2020) the results might have been different.

## 5. Conclusions

Our analysis shows that the rapid adoption of RT guidelines adapted to a severe pandemic situation is associated with compliance higher than 60%. This translates into an overall change of RT treatments in over one third of cases and in a reduction in the delivered RT fractions close to 30%. Further studies in this field could be aimed at (i) understanding the real impact of pandemic-adapted RT guidelines and, in particular, whether they reduce the risk of contagion in addition to the other measures usually proposed in terms of social distancing, hygiene, etc.; (ii) assessing any negative impact in terms of tumor control, quality of life, and toxicity resulting from the adoption of modified RT regimens; (iii) improving the results in terms of compliance through the use of intensified strategies based on weekly monitoring of prescriptions, on the collegial discussion of all cases, and on feedback strategies for the involved ROs; and (iv) performing multi-center analyses aimed at collecting a sufficient number of patients and ROs in order to improve the identification of compliance predictors.

**Author Contributions:** Conceptualization, E.G. and A.G.M.; methodology, C.M. and S.C.; validation, A.G.M.; formal analysis, C.M. and S.C.; investigation, L.S. and A.A.; resources A.Z.; data curation, I.A., A.G. (Alessandra Guido), A.G. (Andrea Galuppi), M.N., G.S., G.T., A.V., and M.F.; writing—original draft preparation, E.G.; writing—review and editing, A.G.M. and M.B.; visualization, E.S. and A.Z.; supervision, S.C.; project administration, A.G.M. All authors have read and agreed to the published version of the manuscript.

**Funding:** This research received no external funding.

**Institutional Review Board Statement:** The study was conducted according to the guidelines of the Declaration of Helsinki and approved by Ethics Committee of IRCCS Azienda Ospedaliero-Universitaria di Bologna (CHRISTIE, CE: 989/2020/Oss/AOUBo).

**Informed Consent Statement:** Informed consent was obtained from all subjects involved in the study.

**Data Availability Statement:** Data supporting reported results can be found at Radiotherapy Unit—A.G. Morganti of IRCCS Azienda Ospedaliero-Universitaria di Bologna.

**Acknowledgments:** We would like to thank Cinzia Giacometti for her invaluable help in data management.

**Conflicts of Interest:** The authors declare no conflict of interest.

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
