# Peer review of "COVID-19 Pandemic-Adapted Radiotherapy Guidelines: Are They Really Followed?"

_curroncol, doi:10.3390/curroncol28050288_

Round 1

Reviewer 1 Report

Garofalo et al present an interesting article on the "real-life" impact of COVID-19 on the activity of a radiation oncology department. The artcile is well written and the conclusions are supported by the results. Methods that have been used are appropriate to the goal of the article. It deserves publication as it is in Current Oncology.

Author Response

We appreciate the encouraging comments. 

Reviewer 2 Report

Well written manuscript.

Author Response

We appreciate the positive feedback from the reviewer.

Reviewer 3 Report

Congratulation for your work and any contribution provided for the completion of the Covid-19 mosaic of knowledge is considered of grate significance.

I feel very grateful with your results because they make us confident for a better future with the involvement of the younger scientists into practice. Their capabilities for being more resilient to unexpected circumstances are very promising. However it would be of great significance to know if the statistical significance in the adaptation regarding the guidelines by the younger scientists is true and thus statistical evaluation has to be completed with the normalization of the results to the age of patient followed and the types of malignancies. We cannot be sure if the results are really true about the age of the scientists and because they had to follow older patients in the majority or certain types of cancer with higher adherence in the guidelines.

Author Response

We would like to thank the reviewer for careful and thorough reading of this manuscript and for the thoughtful comments. The number of doctors in our department is not sufficient to enable us to apply the results obtained on a large scale. To interpret the chi-square table, we performed the residual test, that provides information about what cells contribute to a significant chi-square. Results showed that this significant difference was due to the higher tendency to adhere to guidelines of the younger compared to middle-aged Ros and did not result from the age of the patients or from the site of the disease.

Then, to analyze the relationship between treatment (modified vs not modified) and the other variables analyzed we performed a Binomial Logistic Regression. The Multivariable analysis confirmed only a statistically significant correlation between older patients age and probability of compliance to the adapted guidelines (p = 0.028).

For this reason, we can conclude that the adherence to the adapted guidelines was influenced only by the patient’s age.

A larger sample of ROs would probably have been needed to confirm the result of the chi-square test.

We modify the manuscript according to your thoughtful comment.